Guiding the retraining of convolutional neural networks against adversarial inputs

Durán Francisco 1 2 francisco.duran.lopez@upc.edu
http://orcid.org/0000-0001-9928-133X Martínez-Fernández Silverio 2 silverio.martinez@upc.edu
Felderer Michael 3 4
Franch Xavier 1
1 Universitat Politècnica de Catalunya , Barcelona, Catalunya , Spain
2 Universidad Nacional Autónoma de México , Mexico City , Mexico
3 German Aerospace Center (DLR) , Cologne , Germany
4 University of Cologne , Cologne , Germany
Wan Shibiao
Electronic publication date: 2023 Aug 8
Publication date: 2023
Volume: 9
Electronic Location ID: e1454
Received 2023 Mar 20; Accepted 2023 Jun 6
Copyright: © 2023 Durán et al.
Copyright year: 2023
Copyright holder: Durán et al.
License: This is an open access article distributed under the terms of the Creative Commons Attribution License, which permits unrestricted use, distribution, reproduction and adaptation in any medium and for any purpose provided that it is properly attributed. For attribution, the original author(s), title, publication source (PeerJ Computer Science) and either DOI or URL of the article must be cited.
License URL: https://creativecommons.org/licenses/by/4.0/

Keywords: Neural networks, Software testing, Deep learning, Adversarial inputs, Green AI

Funding: GAISSA Spanish research project ref. TED2021-130923B-I00; MCIN/AEI/10.13039/501100011033 UNAM-DGECI Iniciación a la Investigación: verano-otoño 2021 Universidad Nacional Autónoma de México (UNAM) BEAGAL:18/00064 Austrian Science Fund (FWF) I 4701-N Austrian Research Promotion Agency (FFG) 888127 This work was supported by the GAISSA Spanish research project (ref. TED2021-130923B-I00; MCIN/AEI/10.13039/501100011033), the “UNAM-DGECI: Iniciación a la Investigación (verano otoño 2021)” scholarship provided by Universidad Nacional Autónoma de México (UNAM), the “Beatriz Galindo” Spanish Program BEAGAL18/00064, the Austrian Science Fund (FWF): I 4701-N and the project Continuous Testing in Production (ConTest) funded by the Austrian Research Promotion Agency (FFG): 888127. There was no additional external funding received for this study. The funders had no role in study design, data collection and analysis, decision to publish, or preparation of the manuscript.

==============================
Background

When using deep learning models, one of the most critical vulnerabilities is their exposure to adversarial inputs, which can cause wrong decisions (e.g., incorrect classification of an image) with minor perturbations. To address this vulnerability, it becomes necessary to retrain the affected model against adversarial inputs as part of the software testing process. In order to make this process energy efficient, data scientists need support on which are the best guidance metrics for reducing the adversarial inputs to create and use during testing, as well as optimal dataset configurations.

Aim

We examined six guidance metrics for retraining deep learning models, specifically with convolutional neural network architecture, and three retraining configurations. Our goal is to improve the convolutional neural networks against the attack of adversarial inputs with regard to the accuracy, resource utilization and execution time from the point of view of a data scientist in the context of image classification.

Method

We conducted an empirical study using five datasets for image classification. We explore: (a) the accuracy, resource utilization, and execution time of retraining convolutional neural networks with the guidance of six different guidance metrics (neuron coverage, likelihood-based surprise adequacy, distance-based surprise adequacy, DeepGini, softmax entropy and random), (b) the accuracy and resource utilization of retraining convolutional neural networks with three different configurations (one-step adversarial retraining, adversarial retraining and adversarial fine-tuning).

Results

We reveal that adversarial retraining from original model weights, and by ordering with uncertainty metrics, gives the best model w.r.t. accuracy, resource utilization, and execution time.

Conclusions

Although more studies are necessary, we recommend data scientists use the above configuration and metrics to deal with the vulnerability to adversarial inputs of deep learning models, as they can improve their models against adversarial inputs without using many inputs and without creating numerous adversarial inputs. We also show that dataset size has an important impact on the results.

Introduction

In recent years, deep learning (DL) systems, defined as those software systems with functionalities enabled by at least one DL model, have become widespread due to their outstanding capacity to solve complex problems. The use of DL systems ranges from applications in autonomous driving systems to applications in medical treatments, among others (Stallkamp et al., 2012; Castanyer, Martínez-Fernández & Franch, 2021a; Tran et al., 2021). However, given their statistical and black box nature, it is often the case that DL systems exhibit an unexpected behavior or produce anomalous results when inferring with inputs different to those used in the training stage. We refer as vulnerabilities to such unexpected behaviors or anomalous results, because they can lead to undesirable consequences in real-world DL systems already in production.

The existence of vulnerabilities is commonplace in any type of software system. In traditional systems, i.e., not embedding any DL model, we may find a vast amount of testing approaches to ensure a certain level of robustness (Könighofer & Bloem, 2012). When the tests detect a problem, there are specific solutions to fix the problem or at least mitigate the effects to a large extent (Khan, 2011). In contrast, testing DL systems is completely different, due to their non-deterministic nature: the “correct” answer or “expected” behavior in response to a certain input is often unknown at the time of training the DL model (Barr et al., 2014; Amershi et al., 2019).

One of the most critical vulnerabilities in DL systems is their exposure to adversarial input attacks. Adversarial inputs are inputs that can be misclassified by applying them a perturbation not distinguishable by human eyes. The perturbation is learned by an attacker, which optimizes the inputs to maximize the DL model’s prediction error (Szegedy et al., 2013). Therefore, comprehensive DL system testing approaches need to include techniques to manage adversarial inputs (Machado, Silva & Goldschmidt, 2021). One way to counteract this vulnerability is by retraining the DL models, with the idea that new inputs can be used for training a better model than the original. In this way, selecting the most informative inputs becomes key when retraining.

This work focuses on improving the testing and retraining of DL models in the presence of adversarial inputs. To this end, we apply two different approaches: (i) using the information provided by a number of guidance metrics, which try to identify the most useful inputs according to the criteria of each metric, and (ii) adding adversarial inputs to augment the dataset employing different training inputs from the original training set. Still, questions arise: What is the accuracy, resource utilization, and execution time of using such metrics? What is the accuracy and resource utilization of using a certain retraining configuration? (the execution time of retraining configuration is not included in this question as the number of used inputs determines the computation expense of the retraining). Thus, it is needed to determine specific scenarios in which a particular guidance metric should be employed, as well as optimal retraining configurations. In this context, this work aims at comparing both guidance metrics and configurations for retraining DL models with adversarial inputs based on their accuracy, to improve the DL models’ performance (Vogelsang & Borg, 2019), resource utilization and execution time, both to obtain computational benefits. The results of the research benefit data scientists to test their models’ accuracy against adversarial inputs and to adapt their models to be aware of adversarial inputs, while keeping the trade-off with the resource utilization of the retraining process. We evaluate our method in one typical DL application field, namely image classification. For this reason, we focus on one particular type of DL model, namely convolutional neural networks (CNN), which are widely used for image classification due to their high performance. Since LeCun et al. (1989) introduced them, there has been significant progress in the literature with respect to their accuracy even in challenging datasets, as well as the explanation of their behavior and abstraction (Krizhevsky, Sutskever & Hinton, 2012; Zeiler & Fergus, 2014).

The main contributions of this work are: We conduct a study based on five datasets, in which we evaluate and compare various retraining runs guided by six guidance metrics. We show that using uncertainty metrics for guiding the retraining of CNN models outperforms neuron coverage, surprise-based and random metrics w.r.t. accuracy (i.e., using an augmented test set with adversarial inputs from three different adversarial attacks), resource utilization and execution time of the metrics.

In the same study, we conduct a comparison, in terms of accuracy (i.e., using also an augmented test set with adversarial inputs from three different adversarial attacks) and resource utilization, of three configurations of retraining CNN models. In particular, when doing adversarial retraining from original model weights can achieve results that are aligned with green AI goals.

We provide guidelines to help data scientists to improve their CNN models against adversarial attacks from the perspective of green AI. We recommend retraining CNN models using an augmented training set with adversarial inputs, from original model weights, and the guidance of uncertainty metrics to improve efficiency without diminishing the models’ accuracy.

We provide a replication package, available online (https://doi.org/10.5281/zenodo.7933484).

This document is structured as follows. Section 2 respectively introduces the field of study of our work, the DL architecture used for the models, the guidance metrics, and the adversarial attacks used to create the adversarial inputs. Section 3 describes related work. Section 4 presents the research questions and the study design. Section 5 presents the results. Section 6 presents the discussions. Section 7 describes the threats to validity, and Section 8 draws conclusions and future work.

Background

In this section, we introduce (i) the field of study of our work, namely deep learning (DL) system testing; (ii) the type of DL model we used, namely convolutional neural networks (CNN); (iii) the guidance metrics used in our study, namely: neuron coverage (NC), two surprise adequacy (SA) metrics: likelihood-based surprise adequacy (LSA) and distance-based surprise adequacy (DSA) and two uncertainty metrics: DeepGini (DG) and softmax entropy (SE); (iv) the adversarial attacks used to create adversarial inputs, namely Fast Gradient Sign Method (FGSM), Basic Iterative Method (BIM) and Carlini-Wagner (CW).

Deep learning system testing

Deep learning (DL) system testing has become a relatively new broad research field, in which every year the number of publications related to the subject increases (Zhang et al., 2019; Huang et al., 2020; Martínez-Fernández et al., 2022). Aiming at laying the foundations of software tests to compare DL systems outputs in a consistent and effective way, different frameworks have been created to facilitate the use of these methods (Papernot et al., 2016; Rauber, Brendel & Bethge, 2017; Zhang et al., 2018; Gerasimou et al., 2020; Ahuja, Gotlieb & Spieker, 2022) and likewise a number of testing metrics have been created to compare the properties of these software testing methods (Pei et al., 2017; Ma et al., 2018a; Kim, Feldt & Yoo, 2019; Feng et al., 2020).

Convolutional neural networks

Convolutional neural networks (CNNs) are one of the most used DL architectures in image classification (LeCun et al., 1989; Khan et al., 2020). A CNN consists of an input layer, an output layer, and multiple hidden layers. These hidden layers typically consist of convolutional layers, pooling layers, normalization layers, fully connected layers, or other types of layers to build complex models. Each of the layers contains a different level of abstraction for an image dataset and the weights of the final model are obtained by backpropagation (LeCun et al., 1989, 1998; Guo et al., 2017).

Guidance metrics

Current approaches to DL systems testing evaluate them according to a number of properties, either functional (such as accuracy or precision) or non-functional (such as interpretability, robustness, or efficiency) (Zhang et al., 2019). In order to test the DL system behavior with respect to these properties, it is important to have metrics to compare the behavior of DL models. Among the most used and accepted metrics by the community are those related to neuron coverage, surprise, and uncertainty of new inputs regarding the model (Kim, Feldt & Yoo, 2019; Ma et al., 2021; Pei et al., 2017; Weiss, Chakraborty & Tonella, 2021; Feng et al., 2020).

Neuron coverage (NC)

Pei et al. (2017) proposed the concept of NC in 2017 to measure the coverage of test data of a DL model and to improve the generation of new inputs, arguing that the more neurons are covered, the more network states can be explored, increasing the chances of defect detection (Ma et al., 2018a). The metric is defined as follows. Let D be a trained DL model, composed of a set N of neurons. The neuron coverage NC of an input x with respect to D is given by

(1) NC(x)=|nϵN|activate(n,x)||N|

where activate(n,x) is true if and only if n is activated when passing x to D.

Surprise adequacy (SA)

Kim, Feldt & Yoo (2019) proposed SA in 2019 as the metric that measures the degree of surprise of the model when it confronts new inputs with respect to training inputs, with the hypothesis that a good test set should be “enough” but not too surprising compared to the training set. Kim, Feldt & Yoo (2019) presented two surprise metrics, which we use in our study.

Likelihood-based surprise adequacy (LSA). Let D be a DL model trained on a set T of inputs. The likelihood-based surprise adequacy LSA of the input x with respect to D is given by:

(2) LSA(x)=1|ANL(TD(x))|∑xtϵTD(x)KH(αNL(x)−αNL(xi))

where αNL(x) is the vector that stores the activation values of neurons in the L layer of D when x is entered, TD(x) is the subset of T composed of all the inputs of the same class as x, ANL(TD(x))={αNL(xi)|xiϵTD(x)} and KH is the Gaussian kernel function with bandwidth matrix H (Ma et al., 2021).

Distance-based surprise adequacy (DSA). Based on the distance between vectors representing the neuronal Activation Traces of the given input and the training data (using Euclidean distance). Let D be a DL model trained on a set T of inputs. The distance-based surprise adequacy DSA of the input x with respect to D is given by:

(3) DSA(x)=||αN(x)−αN(xa)||||αN(xa)−αN(xb)||

where:

(4) xa=argminxiϵTD(x)||αN(x)−αN(xi)||

(5) xb=argminxjϵT∖TD(x)||αN(xa)−αN(xi)||

and where D(x) is the predicted class of x by D and αN(x) is the vector of activation values of all neurons of D when confronted with x (Ma et al., 2021).

Uncertainty metrics

The hypothesis of using uncertainty-based metrics as a guidance for retraining is that inputs with higher uncertainty are more likely to be misclassified, they are the most challenging inputs due to their low confidence. Some authors have studied this type of metric and proved that inputs with the highest uncertainty are the most informative for the DL model in a retraining process (Feng et al., 2020; Weiss & Tonella, 2022). In this way, we evaluate uncertainty with two different approaches:

DeepGini (DG). Feng et al. (2020) presented DG, a metric to prioritize massive tests by using just the test input activations of the DL models softmax output layer. In such manner, execution time and memory requirements are only dependent on the number of classes, which makes it highly scalable. Unlike coverage metrics like NC, DG is based on a statistical perspective of DL models, giving higher values to inputs with higher spread of softmax values across classes, which permits to identify possible misclassification of the inputs. Let D be a trained DL model for a classification problem with C classes, the DeepGini DG value of input x with respect to D is given by:

(6) DG(x)=1−∑c=1Clc(x)2

where lc(x) are the softmax values ranging from 0 to 1 for class c and input x, which sum up to 1.

Softmax entropy (SE). This metric is very similar to DG. However, SE is based on the perspective of information theory (Shannon, 1948) and not a statistical perspective, requiring logarithmic computation. Let D be a trained DL model for a classification problem with C classes, the softmax entropy value SE of input x with respect to D is given by:

(7) SE(x)=−∑c=1Clc(x)log⁡lc(x)

where lc(x) are also the softmax values as in DG.

Adversarial attacks

There are DL models whose dataset is augmented when there is not enough data. This can be done through techniques such as obtaining new entries from those that already exist, carrying out transformations to these including cuts and rotations, and synthetic data generation, among others (Tian et al., 2018; Feinman et al., 2017; Jöckel, Kläs & Martínez-Fernández, 2019). Another way is using adversary inputs, which C. Szegedy discovered in 2013 when he noticed that several DL models are vulnerable to slightly different inputs from those that are correctly classified, i.e., the adversarial inputs (Szegedy et al., 2013). This observation caused concerns because it goes against the ability of these models to achieve a good generalization. We use three state-of-the-art adversarial attacks:

Fast gradient sign method (FGSM)

In 2015, Goodfellow, Shlens & Szegedy (2014) introduced the FGSM method to be able to create adversarial inputs in a relatively simple way, forcing the misclassification of the input controlling the disturbance so that it is not perceptible to a human. Given a DL model f with parameters θ, the FGSM is computed as follows:

(8) x∗←x+εsign(∇xJ(f,θ,x))

where J is the cost function used for the training of the model in the neighborhood of the training point x for which the adversary wants to force a wrong classification and ε is a parameter controlling the amount of the perturbation applied to x. The adversary input corresponding to the input x that results from the method is denoted as x∗.

Basic iterative method (BIM)

Kurakin, Goodfellow & Bengio (2018) presented BIM, an iterative version of FGSM. The difference is that in this attack, they apply FGSM multiple times with a small perturbation size, computed as follows:

(9) x0∗←x

(10) xn+1∗←Clipx,ε{xn∗+αsign(∇x(f,θ,xn∗))}

where Clip means clipping the values after each smaller step α to be within an ε-neighborhood of the original input, instead of applying a single step ε.

Carlini-wagner (CW)

The Carlini-Wagner (CW) attack was proposed by Carlini & Wagner (2017). They generate adversarial inputs by finding solutions to an optimization problem. Given a DL model f, with logits layer Z and an input image x with t being the target class, using gradient descent, the Eq. (11) is solved iteratively:

(11) minimize||x−x∗||22+c∙ℓ(x∗)

where the attack searches for a small perturbation x−x∗ that fools the model. A hyperparameter c is used in order to compute the minimal perturbation required for the objective. On top of that, there is a cost function ℓ which operates over the logits Z and the hyperparameter κ to manage the confidence to generate adversarial examples that will be misclassified. However, incrementing it produces perturbations that may be perceptible by humans.

(12) ℓ(x∗)=max(max{Z(x∗)i:i≠t}−Z(x∗)t,−κ)

Related work

From the reviewed literature, there are many studies focusing on the creation of adversarial inputs in order to uncover debilities and vulnerabilities of DL systems or models (Goodfellow, Shlens & Szegedy, 2014; Kurakin, Goodfellow & Bengio, 2018; Carlini & Wagner, 2017). Goodfellow, Shlens & Szegedy (2014) were the first to formulate a concrete definition of the adversarial inputs and clarify their generalization in different architectures and training sets. Additionally, they defined FGSM which allowed generating adversarial inputs. Since then, several adversarial attacks have been proposed (Carlini & Wagner, 2017; Kurakin, Goodfellow & Bengio, 2018; Ren et al., 2020).

Other studies centered on being able to detect such adversarial inputs for different purposes. Feinman et al. (2017) aimed at distinguishing adversarial samples from their normal and noisy counterparts. Ma et al. (2018a) used their coverage criteria to quantify defect detection ability using adversarial inputs created with different adversarial techniques. Kim, Feldt & Yoo (2019) used their proposed SA criteria to show that they could detect adversarial inputs. Wang et al. (2021) considered augmenting the test set with adversarial inputs, to investigate the performance of their test input prioritization approach, PRIMA (PRioritizing test inputs via Intelligent Mutation Analysis), on this type of input.

Although there are several techniques to face the problem of adversarial examples in ML literature (Croce et al., 2020), there is a research challenge focused on using adversarial inputs together with guidance metrics in retraining with the purpose of improving the models (Pei et al., 2017; Ma et al., 2018b; Kim, Feldt & Yoo, 2019), see Table 1 for a summary. In these few works, we can identify in the majority of them, a configuration similar to the adversarial fine-tuning presented in Section 4 of this work, in which the authors retrain with only a few adversarial inputs and do not clearly show the steps for the retraining. Kim, Feldt & Yoo (2019) showed that sampling inputs using SA for retraining can result in higher accuracy, Pei et al. (2017) showed that error-inducing inputs generated by DeepXplore can be used for retraining to improve accuracy. Feng et al. (2020) demonstrated that retraining a model with the prioritization of DG is more effective in improving the DL model accuracy than coverage-based metrics.

Table 1 Review of the retraining configurations used in related work (OAR, AR and AF refer to the configurations presented in Section 4).

Study	Configuration	Independent variables	Studied variables	Datasets	
Pei et al. (2017)	Retraining of an original model with 100 new error-inducing samples (Similar to AF).	DL model	Accuracy	MNIST, ImageNet, Udacity challenge, Contagio/VirusTotal, Drebin	
Tian et al. (2018)	Retraining using synthetic images and original training set. Does not specify if it is done from scratch (Similar to OAR and AR).	Image transformation	Accuracy, MSE	Udacity challenge	
Kim, Feldt & Yoo (2019)	Retraining with 100 new inputs from different ranges of SA metrics (Similar to AF).	Layer of neurons used for SA computation adversarial attack	Accuracy, MSE	MNIST, CIFAR-10, Udacity challenge	
Ma et al. (2021)	As AF, however on each iteration they compute the metrics, instead of calculating metrics once.	Testing metric (Guidance metric)	Accuracy, validation loss	MNIST, Fashion-MNIST, CIFAR-10	
Ma et al. (2018b)	As AF, but when they add a new selected batch, it starts from previous retrained model; not from original model weights.	DL model	Accuracy	MNIST, Fashion-MNIST, CIFAR-10	
Feng et al. (2020)	Retraining using 1%, 2%,…, 10% of an adversarial test set (Similiar to AF).	Test prioritization technique (Guidance metric)	Accuracy, Execution Time, Average Percentage, Faults Detected	MNIST, Fashion-MNIST, CIFAR-10, SVHN	
Weiss & Tonella (2022)	Retraining using 10% or 20% of a half of the test set, using out-of-distribution datasets, which are different to adversarial inputs (Similar to AF).	Test input prioritizer (Guidance metric)	Accuracy, Execution Time, Average Percentage, Faults Detected	MNIST, Fashion-MNIST, CIFAR-10, IMDB	
Guo et al. (2022b)	Similar to AF, using just one adversarial attack.	Acquisition function (Guidance metric)	Accuracy, robustness	MNIST, Fashion-MNIST, CIFAR-10, SVHN	
This study	OAR, AR and AF (see Section 4).	Guidance metric retraining configuration	Accuracy, Resource utilization, Execution Time	GTSRB, Intel, CIFAR-10, MNIST, Fashion-MNIST	

Most recently, Ma et al. (2021) proposed to use testing metrics as a retraining guide, looking for an answer on how to select additional training inputs to improve the accuracy of the model. They used the original training set starting from a previous trained model but ordering these inputs following the metrics’ guidance. Similarly, Weiss & Tonella (2022) used these metrics (Test Input Prioritizers, in their study) to identify the inputs which expose model faults to integrate them to the training set of future retraining runs. They replicated the findings of Feng et al. (2020), however, they found that DG does not in general perform better than other uncertainty metrics and they did not utilize adversarial inputs, but out-of-distribution inputs & inputs from a test set. Compared to these works, we aim to conduct the retraining using the information of an augmented dataset with adversarial inputs and inputs from the original training set.

As the use of adversarial inputs has been proved to provide many improvements, we identified a research gap there to take advantage of them during a retraining process. Moreover, we want to understand how to order an augmented dataset with adversarial inputs guided by metrics so that the retraining is efficient (with highest accuracy and using fewest inputs) for data scientists. To the best of our knowledge, our study is the first one that applies metrics to guide a retraining using an augmented dataset with adversarial inputs in order to improve the model accuracy against adversarial inputs and keeping the trade-off with resource utilization, in order to obtain computational benefits, which is key as the retraining phase is time consuming and the creation of adversarial images has a high computational cost (Shafahi et al., 2019).

In our study, we consider an augmented dataset with adversarial inputs obtained from the original training set using the FGSM, BIM and CW attacks. Therefore our study provides the following novel contributions: A comparison between state-of-the-art guidance metrics for a guided retraining.

A comparison of three different configurations for doing a retraining against adversarial inputs.

Guidelines to help data scientists to improve their CNN models against adversarial attacks from the perspective of green AI.

Empirical study

Research questions

The goal (Basili, Caldiera & Rombach, 1994) of this research is to analyze guidance metrics and retraining configurations for a retraining process of CNN models with the purpose of comparing them with respect to DL testing properties such as accuracy, resource utilization, and execution time from the point of view of a data scientist in the context of image classification.

Thus, we aim at answering the following research questions (RQ): RQ1 - Does the use of guidance metrics impact the accuracy, the resource utilization, and the execution time required for the retraining of a CNN model?

RQ2 - Does the configuration of the retraining of a CNN model impact the accuracy and the resource utilization required for retraining this model?

Variables

Table 2 describes the variables of our study. Regarding the dependent variables, some details follow:

Table 2 The variables of the study.

Class	Name	Description	Values or formula	Scale	
Independent	Guidance metric	Testing metrics used to order the inputs for retraining	NC, DSA, LSA, Random, DG, SE	Nominal	
	Retraining configuration	The configuration used for retraining the models against adversarial inputs	OAR, AR, AF	Nominal	
Dependent	Accuracy	Accuracy using the augmented test set composed of the original test set and an adversarial test set obtained from the previous applying the adversarial attacks to each one of the original test inputs
TP = True Positives; TN = True Negatives FP = False Positives; FN = False Negatives	TP+TNTP+TN+FP+FN	Ratio	
	Resource utilization	u = Input size used to obtain the maximum accuracy
T = Number of total inputs	uT	Ratio	
	Execution time	Time to obtain the metrics values	hh:mm:ss	Interval	
Other	Dataset	The datasets used for training the DL models	GTSRB, Intel, CIFAR-10, MNIST, Fashion-MNIST	Nominal	

Accuracy, measures the capability of the retraining phase of providing higher accuracy against adversarial inputs. We plan to measure the accuracy of the CNN models against an augmented test set, composed of the original test set and an adversarial test set. The latter is obtained from the original test set, applying the chosen adversarial attacks to each input.

Resource utilization, which measures the input size needed to obtain the highest accuracy during the retraining phase.

Execution time, which quantifies the time to compute each of the considered metrics for the corresponding dataset.

Another variable that influences our conclusions is the dataset. We experiment with five different datasets of different size and from different domains. Hence, we define the dataset as a nominal variable indicating the used dataset for training and retraining the models.

Study design

Figure 1 shows the study design. First, the data collection and preprocessing which consists of the collection and preprocessing of raw data. Second, the model training which consists of the traditional training and proposed retraining phase. Finally, we provide answers to the RQs by analyzing our results w.r.t. DL testing properties.

Figure 1 Schema of our empirical study.

Data collection and preprocessing: datasets.

We evaluate the guidance metrics on CNNs using five multi-class, single-image classification datasets.

First, the German Traffic Sign Recognition Benchmark (GTSRB) dataset, with 43 classes containing 39,208 unique images of real traffic signs in Germany. It has been widely used in DL research (Castanyer, Martínez-Fernández & Franch, 2021a; Stallkamp et al., 2012; Loukmane, Graña & Mestari, 2020). This type of images are characterized by large changes of visual appearance for different causes (e.g., weather conditions). While humans can easily classify them, for DL systems (e.g., autonomous driving systems) still is a challenge.

Second, the Intel Image Classification (Intel) dataset with six classes containing 17,034 labeled images of natural scenes around the world, used in many studies as well (Wu et al., 2020; Rahimzadeh et al., 2021; Ren & Li, 2022). It was provided by Intel corporation to create another benchmark in image classification tasks such as scene recognition (Bansal, 2019). This scene recognition task is a daily task for humans and widely used application in industries such as tourism or robotics, still an emerging field for computer vision (Matei, Glavan & Talavera, 2020; Sobti et al., 2021).

Third, the Canadian Institute for Advanced Research (CIFAR-10) dataset, with 10 classes containing 60,000 labeled images of animals and transports, such as dogs, cats, airplanes and ships (Krizhevsky, Nair & Hinton, 2009). This dataset is commonly used to train machine learning and DL models. Compared to MNIST, it is more difficult to train a model due its size and complexity.

Fourth, the Modified National Institute of Standards and Technology (MNIST) dataset, with 10 classes containing 70,000 labeled images of handwritten digits from 0 to 9 which are size-normalized and centered (LeCun & Cortes, 2010). This dataset is also commonly in the training of machine learning and DL models, known for its simplicity in terms of computation.

Fifth, the Fashion-MNIST dataset (Fashion), which considers 10 classes and also contains 70,000 labeled images of clothing, such as T-shirts, pullovers and sneakers (Xiao, Rasul & Vollgraf, 2017). This dataset was provided to be a more challenging alternative to the well-known MNIST dataset.

The adversarial inputs are obtained using the FGSM, BIM and CW attacks from the foolbox library (Rauber, Brendel & Bethge, 2017). Table 3 describes the models we use in our study and the total number of inputs of the augmented test set with adversarial inputs generated by the adversarial attacks.

Table 3 CNN models.

Dataset	# Layers of the CNN (using convolution and dense layers with max-pooling and dropout)	# Neurons	# Original test set inputs	# Augmented test set inputs	
GTSRB	9	1,259	3,923	7,846	
Intel	7	970	3,000	6,000	
CIFAR-10	8	634	6,001	12,002	
MNIST	4	234	7,001	14,002	
Fashion	5	330	7,001	14,002	

Model training

The traditional training of a model consists of using an available original training set identified as “Train” in Fig. 1 and then evaluating the model against an original test set identified by “Test” in Fig. 1. The traditional training is represented in the upper part of Fig. 1 (ii) Model training phase. We plan to add in this phase a retraining process as it is represented in the lower part of Fig. 1 (ii) Model training phase. This retraining uses adversarial inputs guided by the metrics and three different configurations, which are inspired by ML literature (Szegedy et al., 2013; Chen, Wang & Chen, 2020; Jeddi, Shafiee & Wong, 2020), to improve the original model, M, against adversarial inputs. After obtaining a retrained model, M∗, we evaluate the DL testing properties of this retrained model. In our study, we consider and refer to the following weights: Weights from scratch: Random initialization of the model’s weights.

Original model weights (M): Obtained after traditional training, using only inputs from the original training set “Train”.

Retrained model weights ( M∗): Obtained after the retraining process (according to the retraining configuration). These weights are from the 20 data points by selecting the model that achieves the highest accuracy.

The retraining process is shown more extensively in Fig. 2. It comprises the following steps: 1. Step 1. Create adversarial inputs for training and testing: First, we obtain the adversarial training set, “Adv. Train”, by applying the adversarial attacks to a subset of the original training set, “Train”. The number of the adversarial inputs for this case is a small proportion of the entire “Train” set, thus we do not increase the artificial inputs used in retraining and the original input distribution is not diminished either (represented in the upper part of Fig. 3). Putting these two sets together, we obtain the augmented training set, “Train*” (represented in the lower part of Fig. 3). In our experimental setup, each of the three adversarial attacks is applied to only one third of the involved set.

Second, we obtain the adversarial test, “Adv. Test”, set by applying in the same way the adversarial attacks to the entire original test set “Test”. Putting these “Test” and “Adv. Test” sets together, we obtain the adversarial test set “Test*”. Figure 3 provides a comprehensive depiction of this step.

2. Step 2. Compute guidance metrics: Based on the original trained model, M, and the “Train” set, we compute the different metrics for the augmented training set, “Train*” (see Fig. 2 Step 2).

3. Step 3. Order inputs w.r.t. the guidance metrics: According to each metric, we order the inputs for the retraining. With this, we expect that the retrained model, M∗, will be trained first with the images that are more difficult to classify and more informative according to the metrics’ value (see Fig. 2 Step 3).

4. Step 4. Retraining according to the configuration: We implement the retraining in three different ways using ordered inputs according to the following configurations (see Fig. 2 Step 4 and Table 4):

Figure 2 Schema of retraining process.

Figure 3 Schema of obtaining adversarial and augmented sets.

Table 4 Retraining configurations.

Retraining configuration	Initial weights	Final weights	Trained with (inputs are sorted by each guidance metric)	Tested with	
OAR	From scratch	M∗	Train*	Test*	
AR	M	M∗	Train*	Test*	
AF	M	M∗	Adv. Train	Test*	

(a) One-step Adversarial Retraining (OAR): Starting from scratch using the new adversarial inputs and original training set. The model M∗ is retrained with the “Train*” set ordered by highest score of LSA, DSA, DG, SE, NC and Random, respectively, starting from scratch.

(b) Adversarial retraining (AR): Starting from the original model M using the new adversarial inputs and original training set. The model M∗ is also retrained with the “Train*” set as in OAR with the difference that the model M∗ is retrained from the original model M weights.

(c) Adversarial fine-tuning (AF): Starting from the original model M using only the new adversarial inputs. The model M∗ is retrained with the “Adv. Train” set, also ordered by highest score of LSA, DSA, DG, SE, NC and Random, respectively, starting from the original model M weights.

For each configuration, we execute a retraining of the models guided by the six metrics and for each metric we obtain 20 data points as shown in Figs. 4–8. Regarding the hyperparameters configuration, the same configuration is used for the original model and retrained model. Each retraining run for each data point is computed from their respective initial weights. Also, we address randomness of resulted values by executing the retraining randomly, and not through incremental training, which means each data point is obtained by starting the retraining from its initial model weights instead of incremental training (starting from its previous data point).

Figure 4 (A–C) Accuracy of the trained models using GTSRB dataset.

Figure 5 (A–C) Accuracy of the trained models using Intel dataset.

Figure 6 (A–C) Accuracy of the trained models using CIFAR-10 dataset.

Figure 7 (A–C) Accuracy of the trained models using MNIST dataset.

Figure 8 (A–C) Accuracy of the trained models using Fashion-MNIST dataset.

Data analysis

Figures 4–8 correspond to the accuracy of the models against the augmented test set, “Test*”. Table 5 shows the accuracy against the same “Test*” set and resource utilization of the experiments by dataset, configuration and metric used in each case. The “Original accuracy” column shows the accuracy of the original model M against the “Test*” set, which is low due to the adversarial inputs, the “Accuracy w.r.t. augmented test set” column shows the highest accuracy during its respective retraining and the “Resource utilization” column shows the input size to obtain that highest accuracy. Table 6 shows the execution time to compute the considered metrics for every input of the corresponding dataset.

Table 5 Accuracy against augmented test set (original test set and adversarial test set) and resource utilization when the model reach the maximum accuracy for OAR, AR and AF applied to each dataset.

	Accuracy w.r.t. augmented test set	Resource utilization	
Dataset	Config.1	Original accuracy	LSA	DSA	DG	SE	NC	Random	LSA	DSA	DG	SE	NC	Random	
GTSRB	OAR	0.848	0.971	0.968	0.971	0.975	0.968	0.973	35,000/35,287	35,287/35,287	35,000/3,5287	15,750/35,287	35,287/35,287	35,287/35,287	
Intel	OAR	0.687	0.736	0.753	0.756	0.750	0.748	0.749	12,600/14,224	14,224/14,224	12,600/14,224	11,900/14,224	12,600/14,224	13,300/1,4224	
CIFAR-10	OAR	0.577	0.669	0.668	0.664	0.664	0.666	0.659	47,999/47,999	38,400/47,999	38,400/47,999	43,200/47,999	43,200/47,999	38,400/47,999	
MNIST	OAR	0.977	0.987	0.989	0.992	0.990	0.986	0.988	53,200/55,998	25,200/55,998	11,200/55,998	11,200/55,998	55,998/55,998	53,200/55,998	
Fashion	OAR	0.834	0.901	0.907	0.904	0.906	0.893	0.902	55,998/55,998	33,600/55,998	44,800/55,998	44,800/55,998	47,600/55,998	44,800/55,998	
GTSRB	AR	0.848	0.945	0.957	0.958	0.961	0.943	0.940	17,500/35,287	3,500/35,287	8,750/35,287	12,250/35,287	35,287/35,287	29,750/35,287	
Intel	AR	0.687	0.735	0.724	0.730	0.740	0.728	0.738	14,000/14,224	13,300/14,224	13,300/14,224	10,500/14,224	12,600/14,224	14,000/14,224	
CIFAR-10	AR	0.577	0.653	0.660	0.652	0.643	0.636	0.641	47,999/47,999	14,400/47,999	33,600/47,999	26,400/47,999	45,600/47,999	28,800/47,999	
MNIST	AR	0.977	0.987	0.988	0.992	0.991	0.986	0.987	55,998/55,998	22,400/55,998	14,000/55,998	14,000/55,998	55,998/55,998	55,998/55,998	
Fashion	AR	0.834	0.901	0.909	0.906	0.909	0.894	0.892	55,998/55,998	22,400/55,998	28,000/55,998	42,000/55,998	55,998/55,998	36,400/55,998	
GTSRB	AF	0.848	0.958	0.962	0.961	0.960	0.952	0.957	3,610/3,921	3,921/3,921	2,850/3,921	2,850/3,921	3,921/3,921	3,230/3,921	
Intel	AF	0.687	0.721	0.701	0.724	0.720	0.719	0.718	2,850/3,000	2,100/3,000	2,250/3,000	3,000/3,000	2,850/3,000	2,400/3,000	
CIFAR-10	AF	0.577	0.634	0.643	0.640	0.636	0.639	0.653	6,000/6,000	5,700/6,000	6,000/6,000	5,100/6,000	5,400/6,000	5,400/6,000	
MNIST	AF	0.977	0.988	0.988	0.989	0.990	0.986	0.988	6,999/6,999	5,440/6,999	3,060/6,999	3,060/6,999	5,100/6,999	5,780/6,999	
Fashion	AF	0.834	0.904	0.909	0.904	0.907	0.905	0.906	6,460/6,999	6,999/6,999	5,100/6,999	6,120/6,999	6,999/6,999	6,999/6,999	
Notes:

1Configuration of retraining.

Bold numbers indicate the accuracy (at most 0.001% worse than the highest value) and resource utilization for the model with the highest accuracy during retraining.

Table 6 Execution time in hours (hh:mm:ss) to obtain the values of the metrics.

Dataset	LSA	DSA	DG	SE	NC	Random	
GTSRB	00:03:32	00:36:19	00:00:48	00:00:49	05:28:57	00:00:00	
Intel	00:01:11	00:01:47	00:00:09	00:00:09	01:11:02	00:00:00	
CIFAR-10	00:05:54	00:19:47	00:00:25	00:00:27	06:57:19	00:00:00	
MNIST	00:08:51	00:39:02	00:00:22	00:00:23	02:32:26	00:00:00	
Fashion	00:04:38	00:25:10	00:00:12	00:00:12	01:20:43	00:00:00	

In order to compare the impact that guidance metrics and retraining configurations have on the accuracy and resource utilization of the retrained models, we select the best model, according to the accuracy, during the retraining of the 20 data points for each combination of configuration and guidance metric (see marked points on Figs. 4–8 in the Results). We obtain the accuracy and resource utilization of these points and report these values in Table 5 to determine if they can be improved by guiding the retraining with the studied metrics and using the studied retraining configurations. In addition to that, we compare the execution time it takes to compute the guidance metrics in Table 6.

We answer the RQs from two different angles. First, we compare the accuracy changes against the augmented test set in the 20 data points of the retrained models according to each configuration and guidance metric. The augmented test set is composed of the original test set and the adversarial test set created with the same adversarial attacks but using inputs of the original test set; this way we ensure that the test data is never used during retraining. Figure 9 shows samples from the adversarial test set of the used datasets, these are first misclassified and after the retraining process are well classified. Second, we compare the input size required to obtain the model with highest accuracy between those data points according to each configuration and guidance metric.

Figure 9 (A–E) Adversarial input samples.

Captions include: dataset; original model predicted class (incorrectly predicted); retrained model predicted class (correctly predicted).

Results

In this section, we report the results of our empirical study, answering RQ1 and RQ2, and highlighting key takeaways.

To better describe the results, we use Figs. 4–8 and Tables 5 and 6, as explained in the Data analysis section.

Results on guidance metrics (RQ1)

Guidance metrics and accuracy. According to Figs. 4–8, we found in the experiments that uncertainty metrics have the best selection of inputs for the retraining as stated by the observed accuracy. Eleven out of fifteen guided retraining runs obtained the best accuracy using uncertainty metrics (highlighted with bold font in Table 5).

Nearly half of the retraining runs, NC has the worst selection of inputs according to Table 5. In addition to that, NC and LSA need more inputs according to Figs. 4–8 to reach a good accuracy compared to the others.

Mostly, the six guidance metrics reach good accuracy, as the worst difference between the highest and the lowest value is 2.4% of accuracy. However, according to the state-of-the-art accuracy levels and dataset sizes, a minimum percentage of accuracy in DL models can make a great difference in a DL system.

Guidance metrics and resource utilization. Uncertainty metrics can reach a better model with fewer inputs, as much as 11,200/55,998 inputs in the best case with 0.992 of accuracy (MNIST dataset) using less than a third of the available dataset, as shown in Fig. 7A and Table 5. On the other hand, this is not so clear when using Intel dataset, which may be due to the size of the dataset: as there are not enough inputs, the metrics can not make a difference with just a percentage of an already relatively small dataset. Nevertheless, SE was the best option according to Fig. 5B. The best option within SA metrics is using DSA, as it presents good resource utilization. When using NC and random, figures show that it is necessary to use almost the entire dataset.

Overall, uncertainty metrics may be the best due to the fact that they identify the inputs that have a low confidence on their classification. Regarding SA metrics, the model has improved since they first identify the inputs that are most different from the previous training inputs. In that way, the model learns features that it missed during the first training, increasing the correct classifications of adversarial inputs. On the other hand, for NC metric it may be more difficult to identify good inputs for retraining because this metric identifies inputs that cover more neurons according to certain thresholds, which in retraining may not be significant.

Guidance metrics and execution time. We obtained the time to calculate the metrics as stated in Table 6. To obtain NC values it takes more than 3× in terms of execution time w.r.t. the needed to obtain SA values, which may be due to the library used and to a non-optimized metric computation. In addition, the shorter time required to get SA values in comparison with the NC values computation may be also due to the size of the datasets, as it is known that the computation of SA metrics tends to quickly augment as the dataset increases (Weiss et al., 2021). We can observe this trend when comparing the ratio of NC execution time to DSA execution time in each dataset. When we consider Intel dataset, computing NC values takes 39× the DSA execution time, but when using MNIST or Fashion-MNIST dataset, this ratio reduces to almost 3×. Regarding the uncertainty metrics, it is clear that they have a major benefit. It did not take more than a minute to compute these metrics since their computation only depends on the number of classes. This can be observed in Table 6: it takes more time to compute in the case of GTSRB because it has more classes than the other datasets (a total of 43 classes compared to the number of the other datasets, ten classes on CIFAR-10, MNIST and Fashion-MNIST, and only six classes on Intel).

Key takeaways for RQ1: Uncertainty metrics have the best selection of inputs for the retraining as stated by the observed accuracy.

Uncertainty metrics can reach a better model with fewer inputs. DSA also has an acceptable resource utilization.

Uncertainty metrics execution time is insignificant compared to SA and NC metrics. The computation of NC values takes more than 3× in terms of execution time w.r.t. the time needed to obtain SA metrics.

Results on retraining configurations (RQ2)

Configuration of the retraining and accuracy. According to the total number of used inputs, as expected, OAR and AR reached higher values of accuracy than AF. Nevertheless, between these two configurations, the benefits of using AR are greater because with fewer inputs (nearly less than a third in the best cases, see Table 5), it is possible to complete the retraining with high accuracy. Due to the initial weights used in AR, this incremental training creates a bias towards the original training set, which can be the reason why AR does not worsen its accuracy against inputs from the original test set. In the same way, the help of the adversarial training set, “Adv. Train”, which augmented the initial dataset, makes this model better against the adversarial inputs from the augmented test set, “Test*”.

Configuration of the retraining and resource utilization. According to resource utilization, AF would be the best option for the studied configurations. However, it has the lowest accuracies, because of the available input size for retraining. In addition to that, the results from using this configuration present very wavy graphs (see configuration OAR from Figs. 4–8), which makes this configuration unstable and using it may not always lead to the expected results due to random influences. Therefore, Table 5 shows that AF only can be a good option if we aim to execute retraining with just a few inputs, because in all the cases, the accuracy is augmented. Thus, AR can be considered the best option, as it needs fewer inputs than OAR to reach high accuracy.

Key takeaways for RQ2: The benefits of using AR are greater because with fewer inputs, nearly less than a third in the best case, it is possible to retrain the model with high accuracy.

AF can be a good option for retraining the model in those cases in which we prioritize resource utilization.

Discussion

Our experimental results focused on the retraining of models against adversarial inputs, taking into consideration the most used and practical state-of-the-art guidance metrics: LSA, DSA, DG, SE, NC, and Random. On top of that, we considered the manner (i.e., retraining configuration) in which the retraining is done.

As observed in the results, we have verified that NC is not a consistent metric to take into consideration to select inputs when retraining. Our results concur with previous studies, mainly Harel-Canada et al. (2020), which have already stated that NC should not be trusted as a guidance metric for DL testing. On the one hand, NC measures the proportion of neurons activated in a model and it assumes that increasing this proportion improves the quality of a test suit. On the other hand, Harel-Canada et al. (2020) showed that increasing NC value leads to fewer defects detected, fewer natural inputs, and more biased prediction preferences, which is contrary to what we would expect when selecting inputs for a retraining process to have a better model and reduce the input size for that. In our study, the results confirm that NC should not be trusted as a guidance metric. Furthermore, Table 6 shows an evident disadvantage on execution time when computing NC. Overall, different metrics rather than NC should be used when doing guided retraining.

Ma et al. (2021) found that uncertainty-based (variance score and Kullback-Leibler score) and surprise-based metrics (SA metrics, namely DSA and LSA) are the best at selecting retraining inputs and lead to improvements in the number of used inputs, up to twice faster than random selection, in order to find an answer on how to select additional training inputs to improve classification accuracy. Regarding SA metrics, as we did not consider the same uncertainty metrics, we confirm the results obtained in that study: SA metrics, especially DSA, compared to the baseline of random selection and also to NC, are a better option for retraining a DL model w.r.t. accuracy and number of used inputs. For their part, Weiss & Tonella (2022) claimed that SA metrics and NC-based metrics are often outperformed by other uncertainty approaches such as DG or simpler uncertainty metrics including SE, which is not considered in Feng et al. (2020), when retraining a DL model. Regarding uncertainty metrics, we also confirm their results: Uncertainty metrics compared to SA metrics, NC, and random selection, are better and lead to faster improvements. We consider this work as a complement to the results of the RQ about the selection for retraining of Ma et al. (2021), and the same for the similar RQ of Weiss & Tonella (2022), as we focused our work on increment the model’s accuracy against adversarial inputs (see Table 7).

Table 7 Complementary studies.

Study	Results related to our RQ1	Results related to our RQ2	Context	
Ma et al. (2021)	Uncertainty-based metrics, which are different from the ones we used (namely augmented versions of variance score and Kullback-Leibler score), and surprise-based metrics are the best at selecting retraining inputs and lead to improvements up to two times faster than random selection.	Not considered.	Retraining configuration. Their original model is trained with only 10,000 inputs. Then compute the metrics to select a batch of 5,000 inputs from the rest of the training set, and so on, until the entire training set is used.
Adversarial inputs. Adversarial inputs are not considered in the test set to evaluate accuracy, nor for retraining. However, they compute empirical robustness (Moosavi-Dezfooli, Fawzi & Frossard, 2016), which considers adversarial inputs based only on FGSM and randomly picked inputs.	
Weiss & Tonella (2022)	They replicate Feng et al. (2020) results about DG, it outperforms surprise metrics and NC metrics on execution time and active learning effectiveness (accuracy when retraining a model). On top of that, they state that SA metrics, NC-based metrics, and even DG are often outperformed by other simpler uncertainty approaches such as SE or Vanilla Softmax.	Not considered.	Retraining configuration. They split test sets into two parts, one is used for retraining and the other for testing. Then from the original training set and the new inputs added, select the first 20% for retraining.
Adversarial inputs. They use corrupted data instead of adversarial inputs in their test set, which is used for retraining and testing.	

Although our results show that these metrics improve the model faster (in terms of the input size used for retraining), we have found that the size of the dataset greatly affects the results, because when using small datasets it is not so fast. Ma et al. (2021) and Weiss & Tonella (2022) only considered datasets with more than 50,000 available inputs to use in the retraining (the entire training set), but in real-world applications is not always possible to have such large datasets. Unlike these works, we have experimented with a smaller dataset in which the improvements are minor (Intel dataset).

Additionally, an important finding is a variable that we also considered: the configuration of retraining. All the related works have only experimented with their own way to execute the retraining and even sometimes they are not explicit with how they did the retraining process. As we observe in the results, the configuration can change the performance of the model considerably. These results confirm the importance on finding efficient configurations to retrain deep learning models. Considering the studied configurations, using the best configuration studied in this work (AR) can give data scientists a fast and efficient method of retraining.

Regarding the computational complexity of the retraining process, we acknowledge that it increases the complexity of the entire training process. Indeed, a defense strategy against adversarial attacks is needed and a retraining process is one of the most effective strategies (Serban, Poll & Visser, 2020). In that way, according to our results, the least computationally complex process will depend on the application of the model and by following our guidelines it is possible to do a retraining against adversarial inputs without increasing by much the complexity of the retraining. The differences between configurations are the input set used for the retraining and the initial weights, as the same model is used for the retraining process, without increasing the model complexity. In that sense, a lower value of resource utilization with the help of guidance metrics translates into fewer operations.

When using the combination of uncertainty metrics, especially SE metric, and AR retraining configuration, we observe that it can lead to results that are aligned with green AI research, which refers to research that takes into account computational costs to reduce the computational resources spent (Schwartz et al., 2020; Castanyer, Martínez-Fernández & Franch, 2021b), as fewer inputs are needed for retraining and also less time to compute the values of the metric w.r.t. the evaluated options for both independent variables. We observe greater benefits of using AR over the other approaches, as we give greater weight to the efficiency without diminishing the models’ accuracy, and encourage data scientists to build greener DL models.

Previous studies, our results, and new implementations of these metrics (Weiss, Chakraborty & Tonella, 2021; Ouyang et al., 2021; Kim et al., 2020) aim to be a baseline for test methods in DL testing: for data selection in retraining processes, data generation, data selection, etc. In this way, we offer guidelines for data scientists in the context of testing with adversarial inputs during the retraining of CNNs for image classification: Guidance metrics and accuracy. Retraining with the guidance of uncertainty metrics can achieve CNN models with higher accuracy compared with the rest of the studied metrics.

Guidance metrics and resource utilization. When prioritizing the number of inputs used for retraining, uncertainty metrics need fewer inputs to reach higher accuracy.

Guidance metrics and execution time. When considering the execution time to obtain the metrics, uncertainty metrics are by far the best choice w.r.t. to other metrics, with the exception of random.

Retraining configuration and accuracy. When prioritizing accuracy, retraining with original inputs and adversarial inputs from scratch (OAR) is the best option. However, retraining with original inputs and adversarial inputs from original model weights (AR) can reach similar accuracies without considerable differences and this configuration uses fewer inputs.

Retraining configuration and resource utilization. Retraining with original inputs and adversarial inputs from original model weights (AR) has a good resource utilization, in addition to high accuracies w.r.t. the other configurations. Nevertheless, retraining with only adversarial inputs (AF) is the best option when prioritizing only resource utilization.

Green AI. When considering efficiency without diminishing the accuracy of the model, retraining with original inputs and adversarial inputs from original model weights (AR), and the guidance of uncertainty metrics, especially SE, is the best way to retrain a CNN model in the mentioned context.

Threats to validity

In this section, we report the limitations of our empirical study and some mitigation actions taken to reduce them as much as possible.

Regarding construct validity, we include five original models using CNN architecture trained in five different datasets, respectively, in order to mitigate mono-operation bias. Derived from the metrics selected, we use LSA, DSA, DG, SE, NC, and Random metrics to guide the retraining runs, which may have potential threats. However, these metrics have been used in previous work as shown in the Related Works section, lowering this threat. And according to threats related to the configurations, we have reviewed the relevant literature and searched for retraining configurations using adversarial inputs.

Concerning conclusion validity, the quality of the DL models and implementations depends on the experience of the developers. To mitigate this, we provide the implementations organized and following the “Cookiecutter Data Science” project structure (https://drivendata.github.io/cookiecutter-data-science), making them as simple as possible. To increase reliability in our study, we detail the procedure to reproduce our work: the process is shown in the Emperical study section, datasets and replication package are available online. Also, we address the randomness of our results by starting the retraining runs for each data point from their respective initial weights (from original model weights for AR and AF, and from scratch for OAR).

Two threats to internal validity are the implementation of the studied DL models, as well as the computation of the metrics. We used available replication packages from the authors of the metrics, using the same configurations they used for the experiments to minimize this risk. Also, different models are used with different datasets, mitigating that the results of our study of guidance metrics and configurations are caused accidentally.

Threats to external validity stem mainly from the number of datasets, models and adversarial generation algorithms considered. Our results depend also on the datasets, the type of architecture considered, and the device used for training. We believe our results are applicable to image classification datasets. Some of these threats are addressed by the use of five image datasets, several state-of-the-art metrics and three adversarial attacks widely used by the scientific community. Regarding the architecture type, as the adversarial inputs can be generalized across different of types of architectures (Goodfellow, Shlens & Szegedy, 2014) and we use them for the retraining, we also believe that results for other architecture types could be similar to the results for CNN architecture, only further experiments can reduce this threat. Furthermore, our research findings are focused on guiding retraining to address adversarial inputs. Thus, it is possible that our results may not be applicable to other related problems, such as test selection. Additionally, it should be noted that in certain models, metrics derived from softmax may not be an adequate proxy for uncertainty (Gal & Ghahramani, 2016).

Conclusions and future work

In this work, we have studied DL testing metrics for guiding the retraining of models, and we adapted the models’ weights to be aware of adversarial inputs, as the retraining process involves adjusting the weights to better handle them. We performed an empirical study with the metrics and also considered three different configurations of retraining against adversarial inputs and did a comparison of the metrics and the configurations. In summary, we observe that (i) the models are increasing their accuracy against the augmented test set with adversarial inputs as it was sought in the objectives of this work, and (ii) there are computational benefits of using certain metrics and configurations.

The empirical study showed that the uncertainty metrics (such as DG and SE) as guidance for a retraining phase are useful for data scientists when using the following configuration: adversarial retraining from original model weights.

With the previous configuration and metrics, we can improve the accuracy of the models against adversarial inputs by up to 13.3% on the GTSRB dataset, 7.7% on the Intel dataset, 14.4% on the CIFAR-10 dataset, 1.5% on the MNIST dataset and 9.0% on the Fashion-MNIST dataset without the need of using many inputs. Therefore, this can be done by using 34.7%, 73.8%, 30%, 25% and 75% (similar accuracy gain achieved since using 30% unlike Intel dataset) of the available inputs, on the GTSRB, Intel, CIFAR-10, MNIST and Fashion-MNIST datasets, respectively.

Concerning SA metrics, DSA presents competitive values for dependent variables except for the execution time variable. Using random can guide to similar levels of accuracy but when using the recommended configuration, the computational benefits of using uncertainty metrics are that fewer inputs are required as discussed above in percentage, which in turn, translates into fewer adversarial inputs created. Regarding the NC metric, we do not recommend the use of it as it presents the greatest disadvantages on the studied variables. In cases where one-step adversarial retraining from scratch is considered, it can be a time-consuming process as it requires using almost 100% of the inputs to achieve similar accuracy to the recommended configuration.

Additionally, we revealed that the size of the dataset is important when implementing the recommended metrics and configuration. Taking this into account, we need to assess whether it would be worth calculating the metrics when using only small datasets.

In the next step, the reproduction of our experiments on different datasets of varied sizes, unbalanced datasets, and also non-images datasets (Kim & Yoo, 2021) are required to generalize our findings. Moreover, future experiments should study the effect of the dataset size on the studied metrics. Particularly, experiments with other DL architectures are required to confirm or reject that our findings can be used across different architectures. Also, our results should be compared to guided retraining runs using different testing metrics such as new refinements of SA (Kim et al., 2020; Weiss, Chakraborty & Tonella, 2021; Ouyang et al., 2021) or other uncertainty-based metrics (Guo et al., 2022a).

While the results of our study provide meaningful outcomes, there are several dimensions to empirically explore in the future. Thus, it would be relevant to evaluate this type of retraining process against adversarial inputs in the context of continual learning (CL), as there is scarce research (Khan et al., 2022b; Bai et al., 2023; Chou, Huang & Lee, 2022; Khan, Bouaynaya & Rasool, 2022a) in addressing adversarial inputs for CL methods (De Lange et al., 2021). Additionally, it would be valuable to study the effects of adversarial attacks, which utilize distinct algorithms and have different computational complexity (Ren et al., 2020).

Additional Information and Declarations

Competing Interests

Author Contributions

Data Availability

The authors declare that they have no competing interests.

Francisco Durán conceived and designed the experiments, performed the experiments, analyzed the data, performed the computation work, prepared figures and/or tables, authored or reviewed drafts of the article, and approved the final draft.

Silverio Martínez-Fernández conceived and designed the experiments, performed the experiments, analyzed the data, prepared figures and/or tables, authored or reviewed drafts of the article, and approved the final draft.

Michael Felderer conceived and designed the experiments, analyzed the data, authored or reviewed drafts of the article, and approved the final draft.

Xavier Franch conceived and designed the experiments, analyzed the data, authored or reviewed drafts of the article, and approved the final draft.

The following information was supplied regarding data availability:

The code is available at Zenodo: fjdurlop. (2023). fjdurlop/guided-retraining: guided-retraining v1.2.0 (v1.2.0). Zenodo. https://doi.org/10.5281/zenodo.7933484.

Stallkamp, Johannes and Schlipsing, Marc and Salmen, Jan and Igel, Christian. “Man vs. computer: Benchmarking machine learning algorithms for traffic sign recognition.” Neural networks 32 (2012): 323-332. German Traffic Sign Recognition Benchmark (GTSRB) dataset. Retrieved January, 2022, from http://benchmark.ini.rub.de.

P. Bansal (2019). Intel Image Classification (Intel) dataset. Retrieved October, 2022, from https://www.kaggle.com/datasets/puneet6060/intel-image-classification.

Krizhevsky, Alex, and Geoffrey Hinton. “Learning multiple layers of features from tiny images.” (2009): 7. Canadian Institute for Advanced Research (CIFAR-10) dataset. Retrieved October, 2022, from https://www.cs.toronto.edu/~kriz/cifar.html.

LeCun, Y., Cortes, C., & Burges, C. J. (2010). Mnist handwritten digit database (MNIST). Retrieved January, 2022, from http://yann.lecun.com/exdb/mnist/.

Xiao, H., Rasul, K., & Vollgraf, R. (2017). Fashion-mnist: a novel image dataset for benchmarking machine learning algorithms. Retrieved April, 2023, from https://github.com/zalandoresearch/fashion-mnist.

Library used to create adversarial inputs: Rauber, J., Brendel, W., & Bethge, M. (2017). Foolbox: A python toolbox to benchmark the robustness of machine learning models. arXiv preprint arXiv:1707.04131. Code is available at https://github.com/bethgelab/foolbox.

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
