# Peer review of "Guiding the retraining of convolutional neural networks against adversarial inputs"

_PeerJ Computer Science, doi:10.7717/peerj-cs.1454_

## Round 0.1 · original submission · Major Revisions

The reviewers have substantial concerns about this manuscript especially about the dataset size and Step 4 of the method. The authors should provide point-to-point responses to address all the concerns and provide a revised manuscript with the revised parts being marked in different color.

Reviewer 1 ·

Basic reporting

The paper studied deep learning testing metrics for guiding retraining of models. The authors revealed that adversarial retraining from original model weights and performed an empirical study with the metrics and also considered three different configurations of retraining against adversarial inputs. The author's intention is correct, and their efforts deserve encouragement. The paper is well organized. I am convinced of the significance and novelty of the paper's contribution. To meet the increasingly high-quality standard of the journal, I have some points below.

1. What is the basis for the selection of weights?
2. Can weights be designed as an adaptive weight for adversarial retraining?
3. Does adversarial retraining increase computational complexity?
4. For the 4 datasets used, the authors should use at least a few images to show the comparison against retraining.

Experimental design

no comment

Validity of the findings

no comment

Reviewer 2 ·

Basic reporting

In this paper, the authors compare six guidance metrics and three configurations for retraining DL models with adversarial inputs based on their accuracy, resource utilization, and time. The authors conducted experiments and analysis of guidance metrics and configurations on four datasets of different sizes and drew corresponding conclusions. This research is significant in addressing how to efficiently improve the robustness of the model. The paper is well-organized, and the analysis is rigorous and convincing.

Experimental design

(1) The datasets used in this research are relatively simple, and the models used are relatively simple classification models. It is unclear whether these experimental conclusions can be generalized to more complex models and larger datasets.
(2) Step4,AF only considers pre-trained models and adversarial samples. It would be beneficial to explore continual learning training strategies to address the issue of model forgetfulness.

Validity of the findings

Regarding Step4, it is unclear why the OAR's effect is superior to the AR, and whether it is due to hyperparameters configuration.

Additional comments

no comment

Reviewer 3 ·

Basic reporting

In this manuscript, the authors explore the vulnerability of deep learning models, specifically focusing on convolutional neural networks (CNNs), to adversarial inputs within the context of image classification. The primary objective is to enhance the CNNs' robustness against adversarial attacks by evaluating six guidance metrics and three retraining configurations.

Employing an empirical methodology, the researchers scrutinize the accuracy, resource utilization, and execution time associated with retraining CNNs under various scenarios. The findings reveal that adversarial retraining, using original model weights and guided by uncertainty metrics, yields the most optimal performance in terms of accuracy, resource utilization, and execution time.

Based on these results, the authors advise data scientists to adopt the aforementioned configurations and metrics to address the susceptibility of CNNs to adversarial inputs. This approach can effectively improve the model's resilience against adversarial attacks without requiring an excessive number of inputs or generating numerous adversarial instances. Furthermore, the study underscores the crucial influence of dataset size on the outcomes.

Experimental design

The investigators meticulously examine the accuracy, resource utilization, and execution time linked to retraining CNNs in diverse settings. The results demonstrate that adversarial retraining, which incorporates original model weights and is directed by uncertainty metrics, delivers the most favorable performance concerning the accuracy, resource utilization, and execution time.

In Section 4.3.2, Step 1, the generation of the Adv. Train dataset is not clearly explained, leading to confusion. Section 2.4 presents various adversarial methods that produce adversarial samples online, depending on the target model. The distinction between the Train and Adv. Train sets prior to retraining is unclear. Kindly elaborate on the adversarial retraining procedure and provide a more comprehensive explanation.

Validity of the findings

In summary, this manuscript examines the efficacy of diverse guidance metrics in enhancing models' resilience against adversarial inputs. The study underscores the benefits of employing surprise-based (SA) and uncertainty-based metrics as opposed to alternatives like NC.
However, the influence of dataset size on the outcomes is substantial, indicating that future investigations should delve into the effects of varying dataset sizes on the performance of these metrics while pinpointing potential avenues for refinement.

---

## Round 0.2 · accepted · Accept

The authors have addressed all the concerns and I suggest accepting the manuscript.

Reviewer 1 ·

Basic reporting

no comment

Experimental design

no comment

Validity of the findings

no comment

Additional comments

I appreciate authors’ efforts in conducting response to deal with my questions. The authors have basically addressed the points raised by this Reviewer in the revised version.
Recommendation: After formatted, considered for Acceptation.

Reviewer 2 ·

Basic reporting

The authors made their efforts to address the concerns from the reviewers, with more explanations and sufficient experiments. This revised version is acceptable.

Experimental design

None

Validity of the findings

None

Additional comments

None

Reviewer 3 ·

Basic reporting

Thank you for submitting the revised paper. Your responses to my questions were satisfactory, and I do not have any further concerns at this time.
However, upon review, I noticed that there are still some outstanding issues raised by other reviewers that may require further improvement. I would recommend addressing these concerns in order to strengthen the overall quality of the paper.

Experimental design

no comment

Validity of the findings

no comment

Additional comments

no comment